# Diversity of Endophytic Microbes in *Taxus yunnanensis* and Their Potential for Plant Growth Promotion and Taxane Accumulation

**DOI:** 10.3390/microorganisms11071645

**Published:** 2023-06-23

**Authors:** Qiao Liu, Ludan Li, Yujie Chen, Sai Wang, Lina Xue, Weiying Meng, Jihong Jiang, Xiaoying Cao

**Affiliations:** Key Laboratory of Biotechnology for Medicinal Plants of Jiangsu Province, School of Life Sciences, Jiangsu Normal University, Xuzhou 221116, China

**Keywords:** *Taxus yunnanensis*, endophytes, growth-promotion, taxanes

## Abstract

*Taxus* spp. are ancient tree species that have survived from the Quaternary glacier period, and their metabolites, such as taxol, have been used as anticancer drugs globally. Plant–endophytic microbial interaction plays a crucial role in exerting a profound impact on host growth and secondary metabolite synthesis. In this study, high-throughput sequencing was employed to explore endophytic microbial diversity in the roots, stems, and leaves of the *Taxus yunnanensis* (*T. yunnanensis*). The analysis revealed some dominant genera of endophytic bacteria, such as *Pseudomonas*, *Neorhizobium*, *Acidovorax*, and *Flavobacterium,* with *Cladosporium*, *Phyllosticta*, *Fusarium*, and *Codinaeopsis* as prominent endophytic fungi genera. We isolated 108 endophytic bacteria and 27 endophytic fungi from roots, stems, and leaves. In vitro assays were utilized to screen for endophytic bacteria with growth-promoting capabilities, including IAA production, cellulase, siderophore production, protease and ACC deaminase activity, inorganic phosphate solubilization, and nitrogen fixation. Three promising strains, *Kocuria* sp. TRI2-1, *Micromonospora* sp. TSI4-1, and *Sphingomonas* sp. MG-2, were selected based on their superior growth-promotion characteristics. These strains exhibited preferable plant growth promotion when applied to *Arabidopsis thaliana* growth. Fermentation broths of these three strains were also found to significantly promote the accumulation of taxanes in *T. yunnanensis* stem cells, among which strain TSI4-1 demonstrated outstanding increase potentials, with an effective induction of taxol, baccatin III, and 10-DAB contents. After six days of treatment, the contents of these metabolites were 3.28 times, 2.23 times, and 2.17 times the initial amounts, reaching 8720, 331, and 371 ng/g of dry weight of stem cells, respectively. These findings present new insight into the industrialization of taxol production through *Taxus* stem cell fermentation, thereby promoting the conservation of wild *Taxus* resources by maximizing their potential economic benefits.

## 1. Introduction

*Taxus* spp. are endemic, endangered, and first-class protected tree species in China with significant medicinal properties. However, the population of *Taxus* plants has drastically declined due to the Quaternary glaciation, and their seeds have a long dormancy period with a low natural reproduction rate, contributing to a severe depletion of wild resources [1]. The secondary metabolites derived from plants are a valuable natural resource for the development of anticancer drugs, such as taxol, camptothecin, curcumin, betulinic acid, podophyllotoxin, and resveratrol [2,3]. Among these, taxol is a crucial secondary metabolite of *Taxus* plants [4], but its content is notably limited. This has prompted the search for alternative production methods of taxol as a means of preserving the limited *Taxus* resources. Various approaches have been used, including semi-synthesis, cell culture, and chemical synthesis [5,6]. However, these methods are expensive due to the high cost of raw materials and low conversion rates of the synthesis reaction [7,8]. Currently, the major sources of taxol include nursery-cultivated *Taxus* plants, synthetic biology, and endophytic fungi, along with other non-*Taxus* plants. No breakthrough-type progress has been made [9]. Consequently, there is an urgent need to identify alternative, cost-effective methods for taxol production.

Plant endophytes are non-pathogenic microorganisms that reside in the tissues and organs of host plants, exhibiting beneficial effects without causing noticeable infection symptoms [10]. Many endophytes exert their ecological functions by promoting plant growth through nitrogen fixation, phosphate dissolution, siderophore production, and IAA production [11]. Several bacterial strains, such as *Paenibacillus polymyxa* SK1 [12], *Enterobacter cloacae* R7 and *Bacillus cereus* N5 [13], and the *Burkholderia phytofirmans* strain PsJN [14], have shown growth-promoting characteristics with their ACC deaminase activity and efficient IAA production. Borah et al. [15] also isolated endophytic bacteria strains from *Camellia sinensis* (L.) O. Kuntze, with *Lysinibacillus* sp. S24 exhibiting a high phosphate solubilization rate and high IAA production. Furthermore, Singh et al. [16] identified 16 endophytic bacterial strains with effective nitrogen fixation in the sugarcane and other crops. Despite the benefits of plant endophytes in promoting plant growth, limited research has been conducted on isolating endophytic bacteria from *Taxus* plants with similar growth-promoting properties.

In addition to growth-promoting properties, endophytic bacteria also play a very important role in promoting the synthesis of plant secondary metabolites [17]. Studies have found that endophytic bacteria can produce unique secondary metabolites to promote the accumulation of secondary metabolites and their intermediates in plants [18,19,20]. Kilam et al. [21] inoculated *Piriformospora indica* into *Stevia rebaudiana* and found that it could significantly increase the content of stevioside and rebaudioside A. Zheng et al. [22] found that endophytic *Penicillium oxalicum* B4 in *Artemisia annua* can increase the content of artemisinin. Endophytes also promoted the synthesis and accumulation of taxanes in *Taxus* plants. Wang et al. [23] isolated an endophytic *Aspergillus niger* from the bark of the *Taxus chinensis* and prepared it into an elicitor to stimulate the suspension cells of the *Taxus chinensis*. It was found that the taxol yield was the highest when 40 mg elicitor was added, which was more than 2 times higher than that of the control group. Li et al. [24] found that the *T.chinensis* endophytic *Fusarium fermentation* broth could increase the taxol yield of the *Taxus cuspidata* to 2 times the amount. Cao et al. [25] found that the fermentation broth of *Pseudodidymocyrtis lobariellae* KL27 could significantly promote the accumulation of taxol in the needles of the *Taxus chinensis*, which reached 3.26 times the amount in the control after 7 days of treatment. Some studies have also found that endophytic bacteria can affect the accumulation of taxanes in *Taxus* callus [7,26], but the mechanism has not been elucidated. However, most of the current studies on endophytes that can promote the accumulation of taxanes focus on endophytic fungi, and there are few studies on endophytic bacteria and endophytic actinomycetes.

The primary objective of this study was to screen endophytes of the *T. yunnanensis* that could promote growth and the accumulation of taxanes. High-throughput sequencing technology was utilized to analyze the diversity of endophytes in the roots, stems, and leaves of the *T. yunnanensis*, followed by the identification of endophytes with strong growth-promoting abilities by conducting assays for IAA production, nitrogen fixation, inorganic phosphorus, ACC deaminase, and cellulase production. The growth-promoting effect of *Taxus* endophytic bacteria on plants was further confirmed by means of *Arabidopsis thaliana* (*Arabidopsis*) plate and pot experiments. Afterward, the endophytes with good growth-promoting characteristics were utilized to prepare elicitors that were tested on *T. yunnanensis* stem cells, with the secondary metabolic promotion capability of the endophytes being reflected by the taxane content of stem cells.

## 2. Materials and Methods

### 2.1. Plant Sampling

The samples in this study were two-year-old *T. yunnanensis* plants in Tengchong, Yunnan, China (25°03′ N, 98°50′ E). We collected leaves, stems, and roots from nine randomly selected two-year-old *T. yunnanensis* seedlings. In this study, leaf samples were collected from the upper portion of the plant leaves, above the petiole. Stem samples were obtained from lateral branches and were completely separated from the leaves. Root samples were collected from the underground lateral root portion of the plant. The collected samples were washed and dried, then they were sealed with wax and packed into a sterile polyethylene bag. Plant samples were treated immediately when they were transported to the laboratory in an ice cooler.

### 2.2. Illumina-Based Analysis of Endophytic Microbes

Firstly, the surface disinfection was carried out following the method of Schulz et al. [27]. Next, microbial DNA was extracted from the samples using the E.Z.N.A.^®^ Bacterial DNA Kit (Omega Bio-tek, Norcross, GA, USA). The final DNA concentration and purification were determined via a NanoDrop 2000 UV-vis spectrophotometer (Thermo Scientific, Wilmington, DE, USA), and the DNA quality was checked using 1% agarose gel electrophoresis. The database construction and sequencing of high-throughput sequencing samples were completed by Novogene Co., Ltd. (Beijing, China).

The V7-V9 regions of the bacteria 16S rRNA gene of each sample were amplified twice [28], while the fungal amplification region was ITS1-4 [29]. To determine the abundance of communities and sequencing data of each sample, rarefaction curves were plotted for each sample. The Uparse algorithm (Uparse v7.0.1001, http://www.drive5.com/uparse/ accessed on 25 January 2022) [30] was utilized to cluster all effective tags from each sample. The default clustering threshold was set to 97% of the sequence identity, resulting in the formation of Operational Taxonomic Units (OTUs). The OTU sequences were taxonomically annotated using the Mothur method and the SSU rRNA database [31] of SILVA138 (http://www.arb-silva.de/ accessed on 25 January 2022) [32] at a threshold of 0.8–1. Taxonomic information was obtained and the community composition for each sample was analyzed at various classification levels. The representative sequences for all OTUs were aligned using MUSCLE [33] (Version 3.8.31, http://www.drive5.com/muscle/ accessed on 25 January 2022) to establish their phylogenetic relationships. Finally, the data for each sample were normalized using the sample with the smallest amount of data as the standard. Both alpha diversity and beta diversity analyses were performed based on the normalized data. Additionally, the algorithm selected representative sequences for each OTU by selecting the most frequent sequence within the cluster, according to its principles. The Qiime software (Version 1.9.1) was utilized to calculate various diversity indices, including Observed-otus, Chao1, Shannon, Simpson, ace, Goods-coverage, and PD_whole_tree. The Unifrac distance (unweighted Unifrac) was then calculated using the phylogenetic relationship between OTUs [34]. The unweighted Unifrac distance was further constructed using the abundance information of OTUs [35]. Differences between different samples (groups) were analyzed via Non-Metric Multi-Dimensional Scaling (NMDS) and different analyses of the beta diversity index between groups. The function of endophytic bacteria was predicted by PICRUSt [36] and FunGuild [37]. LEfSe analysis uses LEfSe software (Version 1.0), and the default setting LDA Score is 4.

### 2.3. Isolation and Identification of Endophytes

We utilized ultrasonic cleaning to remove any residues from the surface of collected samples, followed by disinfection using the method developed by Schulz et al. [27]. Subsequently, the samples were dried under sterile conditions and ground into king paste, enabling the subsequent isolation of endophytic bacteria and actinomycetes. A gradient dilution approach using sterile distilled water (10^−1^, 10^−2^, and 10^−3^) was employed, and 100 μL of each dilution was evenly spread on various agar media (Luria–Bertani’s, ISP Medium No.2, ISP Medium No.4, Humic acid-Vitamins, and Gause’s Synthetic Agar) and incubated at different temperatures (28 °C, 30 °C, and 37 °C). For the isolation of endophytic fungi, the surface-sterilized dry samples were cut into tissue blocks (approximately 0.5 cm × 0.5 cm), which were attached to a PDA culture and incubated at 28 °C. The resulting colonies were subcultured, purified, and subjected to morphological examination to select colonies exhibiting different phenotypic characters and growth rates for molecular identification and then employed for further studies. Endophytes with different phenotypic characters and growth rates were subjected to molecular identification.

The genomic DNA of both bacteria and fungi were extracted and purified using the rapid bacterial genomic DNA isolation kit (B518225-0100) and the rapid fungal genomic DNA isolation kit (B518229-0100) from Sangon Biotech (Shanghai, China). The DNA quality was checked, and qualified samples were subsequently used for the amplification of 16S rRNA (for bacteria) and ITS (for fungi) sequences. The universal primer sequences for amplifying the 16S rRNA gene of bacteria are 27F (5′-AGAGTTTGATCMTGGCTCAG-3′) and 1492R (5′-TACGGYTACCTTGTTACGACTT-3′). The primer sequences for amplifying the ITS4-5 region of fungi are ITS4 (5′-TCCTCCGCTTATTGATATGC-3′) and ITS5 (5′-GGAAGTAAAAGTCGTAACAAGG-3′). Using the Novozan (Nanjing, China) Kit 2 ×Phanta Max Master Mix’s standard program, they were subjected to PCR reaction with slight modifications. The PCR products were sent to Sangon Biotech (Shanghai, China) for second-generation sequencing.

### 2.4. Growth-Promoting Characteristics of Endophytic Bacteria

The indole-3-acetic acid (IAA) release, siderophore production, phosphate solubilization, 1-aminocyclopropane-1-carboxylic (ACC) deaminase activity, nitrogen fixation, and extracellular protease activity were considered indicators to screen endophytic bacteria with growth-promoting characteristics. The medium used for qualitative and quantitative analysis of IAA production by endophytic bacteria was similar to Sheng et al. [38]. The Schwyn and Neilands [39] ‘universal’ method was used to detect the production of siderophores by endophytic strains. A blue agar plate containing chrome azurol S (CAS) was used and the bacterial strains were spot planted at the center of the medium and then incubated at 30 °C for 5 days. Yellow-orange halos were observed around the colonies, and the diameters of these halos was measured and compared to identify the strains with siderophore production capabilities.

To establish a standard curve for the quantitative analysis of inorganic phosphorus, a KH_2_PO_4_ solution was made by dissolving 0.4388 g of the compound in 80 mL of deionized water and adding 5 mL of concentrated sulfuric acid. The solution was then diluted to 500 mL with deionized water. Next, a 5 mg/L phosphorus stock solution was made by further diluting 5 mL of the above solution in 100 mL of deionized water. Then, concentrations of 0, 1, 2, 4, 6, 8, and 10 mL of the stock solution were added to sterile centrifuge tubes, which were then diluted to 20 mL with deionized water. An indicator solution was added, succeeded by an anticoloring reagent, and the samples were allowed to develop color for 30 min. The OD_700_ value for 200 μL of the resulting solution was measured and the phosphorus concentration and its corresponding OD_700_ value were used to construct a standard curve.

Qualitative and quantitative analysis of inorganic phosphorus by endophytic bacteria was carried out using a PVK medium [40]. After incubation at 30 °C for 5 days, 1 mL of bacterial liquid was collected, and 500 μL of supernatant was volumed to 20 mL with deionized water. After the dinitrophenol indicator was added dropwise, 2 M NaOH was added dropwise until a yellowish color was reached, and then 0.5 M sulfuric acid was added dropwise until a colorless state was reached. Then, 5 mL of molybdenum antimony antireagent was added to the centrifuge tube and volumed to 50 mL with deionized water, then it was shaken well and the OD_700_ value was detected after 30 min. The value was placed into the phosphorus standard curve (y = 0.1917x − 0.0025, R^2^ = 0.9939) to derive the corresponding dissolved phosphorus content.

The ability of endophytic bacteria to produce ACC deaminase was identified using an ADF medium [41]. The strain was incubated on an ADF medium at 30 °C and subjected to three passages to evaluate its proficiency in ACC deaminase production. The nitrogen fixation ability of endophytic bacteria was detected using a JNFbN-free medium (K_2_HPO_4_ 0.5, CaCO_3_ 1.0, MgSO_4_ 0.2, sucrose 10.0, KCl 0.2, agar 20, g/L, pH7–7.5). The tested strain was incubated on a JNFbN-free medium at 30 °C, with three passages, to see the nitrogen fixation ability. The cellulase production ability of endophytic bacteria was detected using a CMC solid medium [42] (K_2_HPO_4_ 0.3, MgSO_4_ 1.0, NaCl 0.5, CMC-Na 10.0, KNO_3_ 1.0, yeast extract paste 2.0, g/L). After incubation at 30 °C for 5 days, 5 mL of 0.5% Congo red solution was added to the plate and left to stand for ten minutes. Then, 1 M NaCl was used to wash the residual Congo red, and the ratio of transparent ring to colony diameter (D/d) was recorded. The detection medium for the protease production ability of endophytic bacteria was as follows: skim milk powder 50.0 g/L, agar powder 20 g/L, pH = 7.2 ± 0.1. The endophytic bacteria were incubated at 30 °C for 5 days, and then the D/d value was recorded.

### 2.5. Plant Experiment

The *Arabidopsis* is a widely used plant model organism for various scientific studies. In this experiment, the efficacy of screened growth-promoting bacteria was further verified using the *Arabidopsis* plant as a test subject. *Arabidopsis* seeds were surface-sterilized for 15 min in 2% (*v*/*v*) hypochlorite and washed five times in sterile water. After vernalization for 12 h in a 4 °C environment, seeds were placed on an MS medium and incubated under artificial illumination. Four-day-old *Arabidopsis* seedlings were submerged in a bacterial suspension for 30 min, washed 3–5 times with sterile water, and placed on a MS medium for growth. The fresh weight, main root length, and lateral root number were measured at 7 and 14 days. Transplanted *Arabidopsis* seedlings were irrigated with 1 mL of a bacterial suspension every other week and monitored for growth status.

### 2.6. Effects of Endophytic Elicitor on the Accumulation of Taxanes in Stem Cells of T. yunnanensis

*T. yunnanensis* stem cells used for an endophytic bacteria induction experiment were cultured in aseptic bottles containing a solid medium of modified B5 [24], which contains 4 g/L of plant gel and 20 g/L of sucrose. The stem cells were cultured at 25 °C in the dark, and subcultured at 2-week intervals on the modified B5 solid medium.

After activation, the tested strains were inoculated into a fresh ISP2 liquid medium at a 2% inoculation amount and cultured at 28 °C, 160 rpm. When the strain grew to a stable stage (stationary phase), the supernatant and bacteria were collected via centrifugation. The supernatant was filtered through a 0.22 μm microporous membrane and stored at room temperature. After freeze-drying, the bacteria were ground into powder and sterilized by moist heat (121 °C, 20 min). The supernatant was added at 4 mL/100 mL of MS medium, and the bacterial powder was added at 4 mg/1 L of MS medium. The *T. yunnanensis* stem cells were cultured for 6 days and then were transferred to the medium supplemented with inducers for another 6 days.

The taxanes, including 10-deacetylbaccatin III, baccatin III, and taxol, were studied via High Performance Liquid Chromatography (HPLC), as described previously [25].

### 2.7. Statistical Analysis

Analysis of variance (ANOVA) and the Student–Newman–Keuls test (*p* < 0.05) were used to compare treatment means. All the statistical analyses were carried out using GraphPad Prism 9.0.

## 3. Results

### 3.1. Endophyte Community Diversity Analysis

Illumina NovaSeq sequencing was utilized to obtain offline data, which went through splicing and quality control to acquire clean tags. Subsequently, filtering of chimeras was performed to obtain effective tags for subsequent analysis. The statistical results of data quality control are shown in Appendix A. The sample dilution curve depicted in Figure 1 illustrated that the sequencing depth of each sample exceeding 40,000 reads (the fungal sequencing depth exceeds 70,000 reads) resulted in a gentle trend of the number of detected species, indicating that the sample sequencing depth reflects the diversity of endophytic microorganisms within the detected samples.

The results of alpha diversity analysis indicated that the richness and diversity of endophytic bacterial communities in the roots of the *T. yunnanensis* were the highest, followed by leaves and stems. Different from endophytic bacteria, endophytic fungi had the highest richness and diversity in leaves, followed by roots and finally stems. The results of the alpha diversity index are shown in Table 1. As shown in Figure 1, the dominant phyla of endophytic bacteria in the one-year-old *T. yunnanensis* plants were Proteobacteria, Bacteroidota, and Actinobacteria. The *Pseudomonas* was the dominant genus in the root, stem, and leaf samples, accounting for 37.28% in the root. In the stem and leaf samples, in addition to the dominant genus *Pseudomonas*, the biological abundance of the remaining genera did not exceed 1% (Appendix A). There were 29 OTUs of bacteria common to each sample, of which *Pseudomonas* accounted for 26.92% (Appendix A). The dominant phyla of endophytic fungi were Ascomycota, Basidiomycota, and Rozellomycota. *Fusarium* was the dominant genus in the root samples, accounting for 22.9%, followed by *Codinaeopsis*, accounting for 15.08%. *Cladosporium* was the dominant genus in stems and leaves, accounting for 47.33% and 37.46%, respectively, followed by *Phyllosticta*, accounting for 19.79% and 26.53%, respectively (Appendix A). A total of 401 strains were common in 3 different parts, and the numbers of unique strains in roots, stems, and leaves were 435, 386 and 491, respectively (Appendix A).

The results of beta diversity analysis showed that the composition of endophytic bacterial communities in the leaves and stems of the *T. yunnanensis* was less different, and they were significantly different from the roots (Appendix A). The LEfSe (LDA effect size) results of roots and stems (Figure 2a,b) indicated that *Pseudomonadaceae*, *Pseuodomonadales*, *Gammaproteobacteria*, *Proteobecteria,* and *Pseudomonas* were the dominant species in stems. The dominant genera in roots were *Chitinophagales*, *Flavobacteriaceae*, *Rhizobiaceae*, *Spingomonadaceae*, *Alphaproteobacteria,* and *Comamonadaceae*. *Pseudomonadaceae*, *Pseudomonadales,* and *Gammaproteobacteria* were the dominant species in leaves. The composition of endophytic fungal communities in the roots, stems, and leaves of the *T. yunnanensis* varies greatly (Appendix A). The results of LEfSe analysis of endophytic fungi (Figure 3a,b) showed that *Sordariomycetes*, *Nectriaceae*, *Hypocreales*, and *Fusarium* were dominant species in roots. *Dothideomycetes*, *Cladosporiaceae*, *Capnodiales,* and *Cladosporium* were the dominant species in the stems. In leaves, *Sordariomycetes*, *Hypocreales*, *Nectriaceae*, and *Fusarium* were the dominant species.

### 3.2. Function Prediction of Endophytes

The overall difference in the functions of endophytic bacteria in the roots, stems, and leaves is relatively small, mainly concentrated in membrane transport, the metabolism of cofactors and vitamins, translation, cellular processes and signaling, lipid metabolism, etc. (Figure 4a). The prediction clustering of endophytic functional genes in roots mainly focused on transporters, ABC transporters, bacterial motility proteins, the secretion system, butanoate metabolism, propanoate metabolism, etc. The functional gene prediction clustering in stems and leaves mainly focused on Arginin and proline metabolism, oxidative phosphorylation, peptidases, methane metabolism, pyrimidine metabolism, and other functions. The functional gene prediction of endophytic fungi mainly focused on plant pathogen soil saprotroph, wood saprotroph, plant pathogens, fungal parasites, Lichenization, and so on (Figure 4b).

### 3.3. Isolation of Culturable Endophytes

In this study, 135 endophytes were isolated from the *T. yunnanensis*, including 108 endophytic bacteria and 27 endophytic fungi. The endophytic bacteria isolated from roots, stems, and leaves were 45, 36, and 27, respectively. The selected 65 strains of endophytic bacteria with obvious characteristics belonged to 4 phyla, 17 families, and 23 genera. The numbers of strains belonging to Firmicutes, Proteobacteria, Actinobacteria, and Basidiomycota were 31, 17, 16, and 1, respectively. The dominant genera of Proteobacteria strains were *Sphingomonas*, *Moraxella,* and *Brevundimonas*, and the quantities were five, two, and three, respectively. The others were *Rhizobium*, *Methylobacterium*, *Marseille*, *Burkholderia*, *Naxibacterium*, *Roseomonas,* and *Paracoccus*. Among the Firmicutes strains, *Bacillus* accounted for 77.42%, *Lysinibacillus* accounted for 3.23%, *Paenibacillus* accounted for 12.90%, and *Staphylococcus* accounted for 6.45%. Actinobacteria strains are mainly composed of three *Micrococcus*, three *Kocuria*, three *Micromonospora*, two *Microbacterium*, two *Janibacter*, one *Blastococcus*, one *Brevibacterium*, and one *Curtobacterium*. The strain TRI2-20 in Basidiomycota belongs to *Erythrobasidium*.

### 3.4. Growth-Promoting Characteristics of Endophytic Bacteria

The present study identified a diverse range of endophytic bacteria with various functional capabilities in different parts of the plant. Specifically, the screening process revealed that among the isolated bacteria, 57 strains were capable of producing Indole-3-acetic acid (IAA), 32 strains of solubilizing inorganic phosphorus, 47 strains of nitrogen fixation, 35 strains of producing cellulase, 30 strains of producing siderophore, 17 strains of producing protease, and 19 strains of ACC deaminase production.

The results showed that there were 18 strains with IAA production greater than 3 mg/L and 5 strains with IAA production greater than 10 mg/L, including the *Sphingomonas* sp., *Staphylococcus* sp., *Bacillus* sp., *Micrococcus* sp., and *Kocuria* sp. Among them, the IAA production of *Sphingomonas* sp. MG-2 was the highest, up to 32.16 mg/L (Figure 5a,b). The results showed that there were 14 strains with dissolved phosphorus content greater than 150 mg/L and 4 strains with dissolved phosphorus content greater than 400 mg/L, which were *Micromonospora* sp. TSI4-1, *Staphylococcus* sp. TLI2-4, *Bacillus* sp. TRI2-16, and *Kocuria* sp. TRI2-1, respectively. Among them, the dissolved phosphorus content of TLI2-4 and TSI4-1 exceeded 500 mg/L, reaching 528.78 mg/L and 530.52 mg/L, respectively (Figure 5c,d).

After 5 subcultures, 19 endophytic bacteria can still grow normally on the medium producing ACC deaminase, and these endophytic bacteria have the potential to produce ACC deaminase. Figure 5f shows the growth of strains TSI2-4 and TSI2-6 after five subcultures. Figure 5e shows the growth of strains TRI4-1 and MG-2 after five subcultures.

The ratio of halo diameter (D) to colony diameter (d) D/d reflects the ability of strains to produce siderophores, and the D/d value of 21 strains was greater than 1.6 (Figure 6a,b). The D/d values of seven strains were greater than or equal to 2, including strains TI2-2, TRI2-1, TI2-5, TI2-3, TSHV-4, TLI2-9, and TLI2-10. Among them, the D/d of *Staphylococcus* sp. TLI2-9 and *Bacillus* sp. TLI2-10 exceeded 3, reaching 3.67 and 3.18, respectively. The ratio of transparent circle diameter (D) to colony diameter (d) D/d reflects the level of cellulase activity of the strains, and the D/d value of 22 strains was greater than 4 (Figure 6c,d). The D/d value of 12 strains was greater than 6, showing a strong ability to produce cellulase, including TSI4-1, TLG1-2, TLG1-5, TLG1-7, TLI2-1, TLI4-2, TSI2-3, TSI2-4, TRI2-12, TRI2-13, TRI2-16, and TRI2-14. The D/d value of *Micromonospora* sp. TSI4-1 was the highest, reaching 9.33.

In the protease detection experiment, 17 strains produced transparent circles around the colonies (Figure 6e,f). Among them, the D/d values of five strains were greater than 3, and the D/d value of TI2-5 reached 7.56.

### 3.5. Effects of MG-2, TRI2-1, and TSI4-1 on the Growth of Arabidopsis

Based on the experimental results of growth-promoting characteristics, three endophytic bacteria (*Sphingomonas* sp. MG-2, *Kocuria* sp. TRI2-1, and *Stenotrophomonas* sp. TSI4-1) of the *T. yunnanensis* were screened for the growth-promoting *Arabidopsis* research. Their phylogenetic trees show a distinct clustering of these three endophytic isolates (Figure 7).

Figure 8a–e depicts the growth and growth indices of *Arabidopsis* on the plate after seven days of endophytic bacteria treatment. The results show that the treatment with strain MG-2 led to a substantial increase in the main root length (86.34%) and number of lateral roots (125%), as well as an 8.25% increase in fresh weight, as compared to the control group. Similarly, after treatment with strain TRI2-1, the main root length and number of lateral roots increased by 83.56% and 48.67%, respectively, and the fresh weight increased by 37.06%. Moreover, treatment with strain TSI4-1 resulted in remarkable improvements, with a 185.84% increase in the number of lateral roots and an 8.59% increase in fresh weight.

In Figure 8f–j, the growth indicators of the *Arabidopsis* on the plate after 14 days of endophytic bacteria treatment are presented. The results demonstrate that treatment with strain MG-2 led to a significant increase in the number of lateral roots (157.39%) and fresh weight (139.7%) compared to the control group. In contrast, treatment with strain TRI2-1 did not significantly alter the main root length or fresh weight. Additionally, the number of lateral roots was slightly lower than that of the control group. Treatment with strain TSI4-1 resulted in a significant increase in the fresh weight of lateral roots by 69.72% and 170.28% respectively.

After transplanting the *Arabidopsis* into pots for 14 days, almost all of the plants treated with the three endophytic bacterial strains had exhibited bolting. Among these, the plant group treated with strain TRI2-1 had the fastest bolting, while the bolting heights of plants treated with strains TRI2-1 and MG-2 were similar (Figure 8k). After transplantation for 28 days, the stem length in the treatment group was higher than that in the control group, with the experimental group exhibiting stem length increases of 1.45, 1.37, and 1.26 times that of the control group, for strains TRI2-1, TSI4-1, and MG-2, respectively. Furthermore, when transplanted for seven days, the rosette leaves in the strain treatment group were significantly larger than those in the control group. After treatment with strains TRI2-1, TSI4-1, and MG-2, the fresh weight increased by 2.13, 1.90, and 1.88 times, respectively, compared to the control group. At 14 days after transplantation, plants treated with strain TRI2-1 showed the most significant increase in fresh weight, at 2.03 times that of the control group. By 28 days after transplantation, the respective effects of the three strains on the A. thaliana fresh weight were ranked in descending order as MG-2, TRI2-1, and TSI4-1, with fresh weight increases of 31.54%, 26.72%, and 18.04%, respectively.

### 3.6. The Effects of MG-2, TRI2-1, and TSI4-1 on the Accumulation of Taxanes in T. yunnanensis Stem Cells

After treatment with strain MG-2, the levels of taxanes were found to be lower than those in the control group (CK). However, the levels of taxanes were significantly higher after treatment with the fermentation filtrate of strain MG-2. Notably, the taxol content was 1.77 times higher than that of the CK group, while the baccatin III and 10-DAB contents were 1.53 and 1.68 times higher, respectively (Figure 9a–c).

After treatment with strain TSI4-1, the levels of taxanes were found to be higher than those in the control group (CK). The taxol content was 1.31 times higher than that of the CK group, while the baccatin III and 10-DAB contents were 1.19 and 1.29 times higher, respectively. Treatment with the fermentation filtrate of strain TSI4-1 resulted in a significant increase in the content of taxanes, with the taxol content being 3.28 times higher than that of the CK group, while the baccatin III and 10-DAB contents were 2.23 and 2.17 times higher, respectively (Figure 9d–f).

After treatment with strain TRI2-1, the levels of taxanes were found to be lower than those in the control group (CK). However, the levels of taxanes were significantly higher after treatment with the fermentation filtrate of strain TRI2-1. The taxol content was 1.71 times higher than that of the CK group, while the baccatin III and 10-DAB contents were 1.61 and 1.53 times higher, respectively (Figure 9g–i). The changes in the content of taxanes were similar to those observed after treatment with strain MG-2.

## 4. Discussion

In this research study, we have conducted an analysis of the diversity of endophytes in the roots, stems, and leaves of 1-year-old *T. yunnanensis* plants, utilizing high-throughput sequencing technology. Our findings have indicated that, amongst the endophytic bacteria found, *Pseudomonas* of Proteobacteria was the predominant genus. Additionally, *Fusarium*, *Codinaeopsis*, *Cladosporium*, and *Phyllosticta* occurred as the dominant endophytic fungi identified. An analysis of the endophytic bacteria isolated through this study fell mainly under 3 phyla, namely Firmicutes, Proteobacteria, and Actinobacteria, with *Bacillus* emerging as the dominant genus, with a total of 31 strains. In accordance with the results of this research study, Proteobacteria has been noted as the predominant genus in numerous plant endophytes, as found in prior studies [43]. For instance, it has been reported to account for 56–74% of the total bacterial population in vine tissue that is free from canker [44]. Similarly, up to 98% of sugar beet (*Beta vulgaris* L.) endophytes belong to the *Proteobacteria phylum* [45]. Furthermore, an increase in CO_2_ concentration in wheat resulted in an elevation of *Pseudomonas* proportion from 82% to 90.32% [46]. Proteobacteria endophytes have demonstrated remarkable effects in respect of plant stress resistance, growth promotion, soil modification, etc. [11,47,48]. These findings present a reference for the potential screening of growth-promoting endophytes in the latter stage of this study.

Reportedly, over 800 genera of endophytic fungi have been recorded worldwide, with the dominant genera including *Alternaria*, *Aspergillus*, *Colletotrichum*, *Fusarium*, *Penicillium,* and *Phoma* [49]. These fungi have mainly been isolated from angiosperms and conifers. Approximately 35% or more of the research regarding endophytic fungi focuses on the leaves of the host [49]. This could be due to plant leaves being more easily colonized by endophytic fungi than other tissues, which aligns with the results of the highest endophytic fungi diversity identified in the leaves of this study. However, with the exception of *Fusarium*, the other dominant genera were not particularly prominent in this study, possibly suggesting differences between hosts. In conjunction with the outcomes of the amplicon data analysis, the isolated endophytic bacteria accounted for only a minor proportion of the total endophytic bacteria of *T. yunnanensis*. In nearly all endophyte isolation studies, culturable microorganisms merely account for a small fraction of the overall microorganisms [50]. Thus, it is necessary to enhance the purification and separation methods of endophytes to gain a better understanding of the endophytic resource of *T. yunnanensis*.

Research studies have identified plant endophytes as microorganisms capable of obtaining nutrients from the soil, transferring these nutrients to plants through root-feeding cycling and other symbiotic nutrient transfer mechanisms, and secreting secondary metabolites for promoting plant growth and development [10]. Moreover, as the second genome of plants, endophytic bacteria can synthesize and secrete an array of plant hormones, antioxidant enzymes, siderophores, volatile organic compounds, and ROS-scavenging enzymes to regulate the expression of plant genes, subsequently impacting plant growth [51].

For instance, the *Pseudomonas aeruginosa* L10 genome has been found to contain a gene cluster, *rhlABRI*, related to rhamnolipid biosynthesis, two sets of genes involved in siderophore biosynthesis (*pvcABCD* and *pchABCDREFG*), and tryptophan biosynthesis genes that aid in IAA biosynthesis (*trpAB*, *trpDC*, *trpE*, *trpF*, and *trpG*) [52]. The predictions of the endophytic bacteria’s functional roles in this study also indicated their involvement in the synthesis and secretion of secondary metabolites, such as membrane transport, amino acid metabolism, carbohydrate metabolism, energy metabolism, cofactor and vitamin metabolism, signal transduction, and so on. The predicted functions reveal the presence of numerous endophytic bacteria may be capable of promoting plant growth and enhancing the synthesis of plant secondary metabolites.

Endophytes are known to be rich in secondary metabolites that play a crucial role in regulatory mechanisms for microorganisms and plant interactions, including phloroglucinols [47], phenazine [53], biosurfactants [54], and hydrogen cyanide [55]. Besides secondary metabolites, endophytes also have the ability to directly or indirectly modify the microenvironment in and around plant tissue, ultimately affecting plant growth. Endophytes have been found to promote plant growth through various mechanisms, such as combined nitrogen fixation, phosphorus dissolution, the production of siderophores, and the synthesis of plant hormones and ACC deaminase [43]. For example, *Paenibacillus polymyxa* SK1 isolated from *Lilium lancifolium* showed the characteristics of promoting plant growth, such as producing organic acids, ACC deaminase, IAA, siderophore, nitrogen fixation, and dissolved phosphate [12]. In recent years, a large number of endophytic bacteria and actinomycetes with growth-promoting characteristics have also been isolated from plants such as corn [56], burdock [57], black pepper [58], and the *Camellia sinensis* [59]. In the realm of endophytic microorganisms, research on the plant growth promotion capabilities of endophytic actinomycetes has been substantially less extensive in comparison to endophytic bacteria [60]. Within the domain of endophytic actinomycetes, there has been more research focus on examining the growth promotion properties of *Streptomyces*, while other genera of actinomycetes have received less attention in research [61]. Therefore, in this study, representative strains were selected from the endophytic actinomycetes of the *T. yunnanensis* to investigate growth-promoting potential. Subsequent screening led to the discovery that a significant number of endophytic bacteria from the *T. yunnanensis* possessed growth-promoting abilities, with several producing multiple growth-promoting factors that could act through both direct and indirect mechanisms.

This study identified two actinobacteria (*Kocuria* sp. TRI2-1, *Micromonospora* sp. TSI4-1) and one bacteria (*Sphingomonas* sp. MG-2) with exceptional growth-promoting potential, validating their practical application in plant development. The effects of MG-2, TRI2-1, and TSI4-1 on the main root length, fresh weight, and lateral root number of *Arabidopsis* aseptic seedlings were evaluated after 7 and 14 days of treatment, resulting in a notable increase in growth. Furthermore, the growth-promoting abilities of the three strains were also observed in *Arabidopsis* plants grown in a soil environment after root irrigation. Previous studies have highlighted the growth-enhancing properties of specific bacterial strains, such as *Bacillus megaterium* RmBm31, isolated from the perennial legume Retama monosperma [62]. This strain was found to activate signaling pathways involving indole-3-acetic acid (IAA), abscisic acid (ABA), and jasmonic acid (JA) to stimulate growth in *Arabidopsis*. Similarly, a newly discovered endophytic bacterium, *B. platensis*, isolated from *Glyceria chinensis*, has also been shown to promote the growth of the *Arabidopsis* [63]. The findings presented above indicate that the mechanisms by which endophytes promote plant growth may involve a range of genes related to growth promotion and nutrient utilization. This underscores the significance of exploring the specific growth-promoting mechanisms of the three endophytic bacteria identified in the latter stages of this study.

In addition to their role in promoting plant growth, endophytes have been found to promote the accumulation of secondary metabolites in plants. For instance, the endophytic fungus *Penicillium oxalicum* B4 has been found to stimulate the biosynthesis of artemisinin in *Artemisia annua* by activating oxidative stress [22]. Similarly, the inoculation of grapevine endophytes *Enterobacter ludwigii* EnVs6, *Pantoea vagans* PaVv7, and *Sphingomonas phyllosphaerae* SpVs6 has been found to increase the content of vanillic acid in grape seedlings [20]. In this context, this study aimed to select endophytic bacteria that could simultaneously promote plant growth and the accumulation of secondary metabolites, specifically taxanes in *T. yunnanensis* stem cells. This study found that the fermentation filtrate of strain TSI4-1 had the most significant effect on the accumulation of taxanes, with taxol content reaching 8.72 mg/g of stem cells (dry weight) after induction. Following a five-day induction period utilizing a *Rhyzopus stelonifera* inducer, *Taxus baccata* stem cells demonstrated a significant increase in paclitaxel content [64]. The paclitaxel concentration reached 25.16 mg/L, which was 16-fold higher than the control group. However, the mechanism of this increase has not been elucidated.

Other studies have shown that endophytic bacteria can increase the content of secondary metabolites through the increased activity of key enzymes in the biosynthesis pathways. For instance, *Novosphingobium resinovorum*, *Rhizobium radiorhizobium*, *Pseudomonas thivervalensis*, and *Pseudomonas frederiksbergensis* increase the activity of 3-hydroxy-3-methylglutaric acid 1-CoA reductase and 1-deoxy-d-xylulose-5-phosphate synthase in the tanshinone biosynthesis pathway, leading to a significant increase in tanshinone content in *Salvia miltiorrhiza* hairy roots [19]. Similarly, *Pseudomonas fluorescens* ALEB7B synthesizes and secretes indole-3-acetic acid to enhance the accumulation of sesquiterpenes in plants [65]. In this study, the fermentation broth of the selected three endophytic bacteria was found to contain elicitors that induced the expression of key enzyme genes in the taxol biosynthesis pathway. However, the structure of these elicitors needs to be further determined by LC-MS and NMR Spectroscopy. Interestingly, the taxane content in stem cells decreased after treatment with strains MG-2 and TRI2-1 compared to the control group. This decrease may be due to the inhibition of the expression of key enzyme genes or the degradation of taxanes by substances secreted by the two strains.

## 5. Conclusions

In conclusion, the abundance of endophytic bacteria in the roots, stems, and leaves of the *T. yunnanensis* has the potential to enhance plant growth. Following screening, *Micromonospora* sp. TSI4-1, *Sphingomonas* sp. MG-2, and *Kocuria* sp. TRI2-1 were identified as having strong growth-promoting functions and the ability to increase taxane accumulation in *T. yunnanensis* stem cells. Further investigation is required to uncover the mechanisms of promoting taxol accumulation. These endophytic bacteria offer potential for large-scale taxol production and have broad applications in biotechnology. Ultimately, this study highlights the significance of endophytic bacteria as a source of bioactive compounds and their potential for sustainable plant bioprocessing and crop improvement. Future research should continue to explore their potential for promoting plant growth and improving the yield and quality of medicinal plant products and elucidate the relevant molecular mechanisms.

## Figures and Tables

**Figure 1 microorganisms-11-01645-f001:**
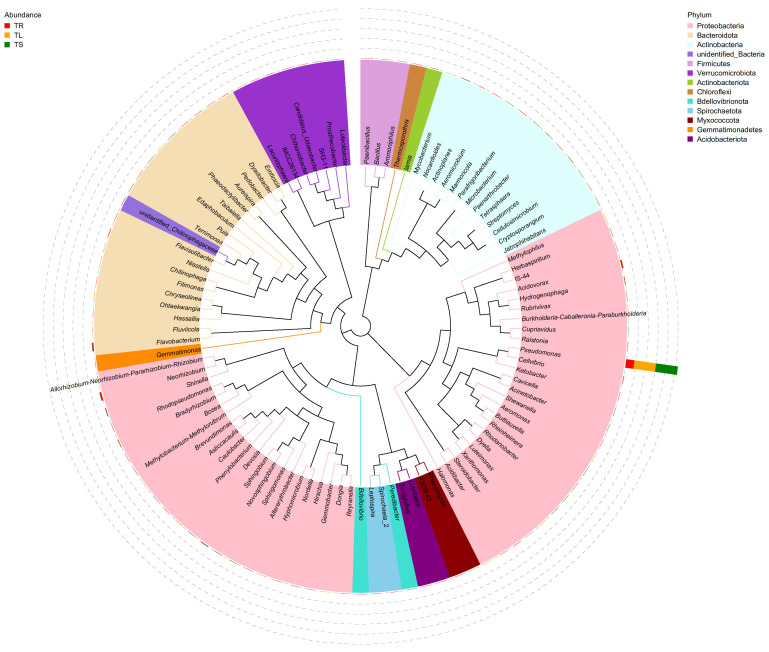
The phylogenetic tree of endophytic bacteria at the genus level. The representative sequences of the top 100 genera were obtained through multiple sequence alignment, and the phylogenetic tree was constructed based on the horizontal species of each genus. The colors of the branches and fan rings correspond to their respective clades. The stacked histograms outside the fan rings convey the abundance distribution information for each genus across various samples.

**Figure 2 microorganisms-11-01645-f002:**
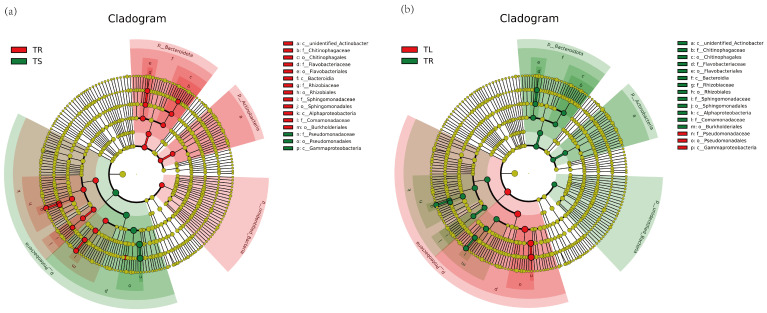
LEfSe analysis of endophytic bacteria in *Taxus yunnanensis*: (**a**) TR vs. TS; (**b**) TL vs. TR. In the evolutionary branch diagram, the circles radiating from inside to outside represent the classification level from phylum to genus (or species). Each small circle at different classification levels represents a classification at this level, and the diameter of the small circle is proportional to the relative abundance. Coloring principle: The species with no significant differences were uniformly colored as yellow, and the biomarkers of the different species followed the group for coloring. The red node represents the microbial group that plays an important role in the red group, and the green node represents the microbial group that plays an important role in the green group. If a group is missing in the figure, it indicates that there are no significant differences in this group. The species names represented by the English letters in the figure are displayed in the right legend. TR, root. TL, leaf. TS, stem.

**Figure 3 microorganisms-11-01645-f003:**
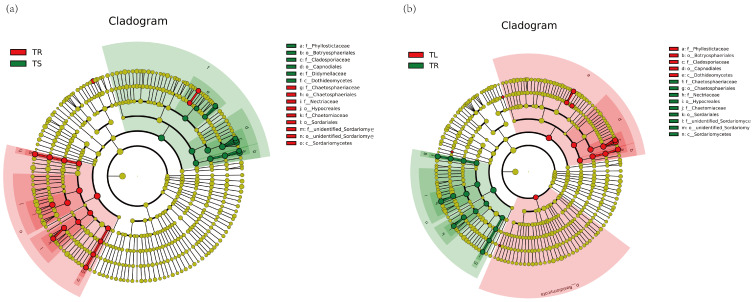
LEfSe analysis of endophytic fungi in *Taxus yunnanensis*: (**a**) TR vs. TS; (**b**) TL vs. TR. Each small circle at different classification levels represents a classification at this level, and the diameter of the small circle is proportional to the relative abundance. Coloring principle: The species with no significant differences were uniformly colored as yellow, and the biomarkers of the different species followed the group for coloring. The red node represents the microbial group that plays an important role in the red group, and the green node represents the microbial group that plays an important role in the green group. TR, root. TL, leaf. TS, stem.

**Figure 4 microorganisms-11-01645-f004:**
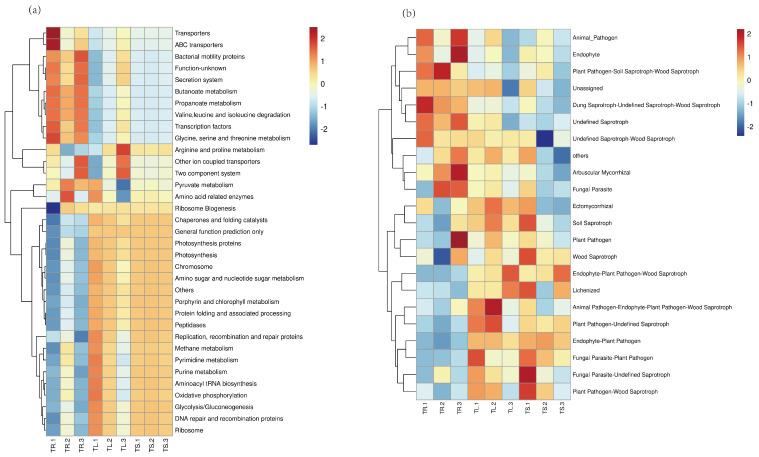
Cluster heat map of PICRUSt functional annotation of endophytes in *Taxus yunnanensis*: (**a**) endophytic bacteria; (**b**) endophytic fungi.

**Figure 5 microorganisms-11-01645-f005:**
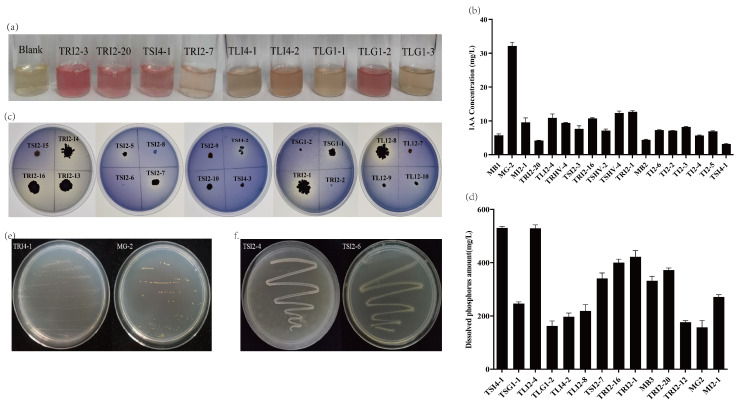
The results of IAA production, phosphorus solubilization ability, nitrogen fixation ability, and ACC deaminase production ability of endophytic bacteria in *Taxus yunnanensis*: (**a**) IAA qualitative experiment, partial results; (**b**) quantitative experiment of 18 strains with strong IAA production capacity; (**c**) phosphate dissolving capacity qualitative experiment, partial results; (**d**) quantitative experiment of 14 strains with strong phosphate dissolving capacity; (**e**) potential of endophytic bacteria producing ACC deaminase of strains TRI4-1 and MG-2; (**f**) nitrogen fixation potential of strains TSI2-4 and TSI2-6.

**Figure 6 microorganisms-11-01645-f006:**
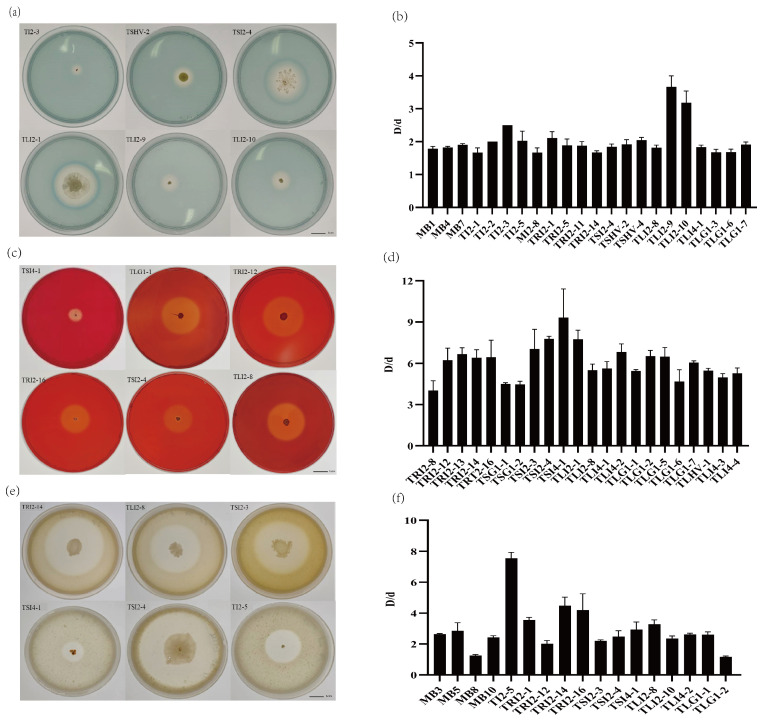
Qualitative experiments of endophytes producing siderophore, cellulase, and protease: (**a**) endophytes producing siderophore plate experiment; (**b**) the ratio of transparent circle diameter (D) to colony diameter (d) in iron carrier plate experiment; (**c**) endophytes producing cellulase plate experiment; (**d**) the ratio of transparent circle diameter to colony diameter in cellulase-producing plate experiment; (**e**) endophyte protease plate experiment; (**f**) the ratio of transparent circle diameter to colony diameter in protease-producing plate experiment.

**Figure 7 microorganisms-11-01645-f007:**
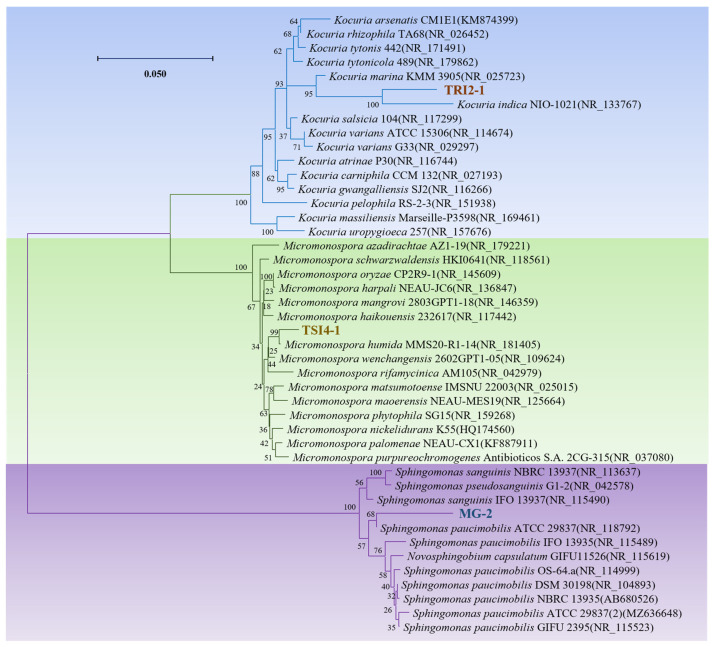
Phylogenetic tree of *Sphingomonas* sp. MG-2, *Kocuria* sp. TRI2-1, and *Micromonospora* sp. TSI4-1. The phylogenetic tree was drawn using the full-length sequence of 16S rRNA gene. MEGA X was used to construct phylogenetic trees based on the alignment of all sequences using the ClusterW method and subsequent trimming of the aligned sequences back and forth. The maximum likelihood method was applied to construct the trees, with 1000 bootstrap replications selected and default values used for other parameters.

**Figure 8 microorganisms-11-01645-f008:**
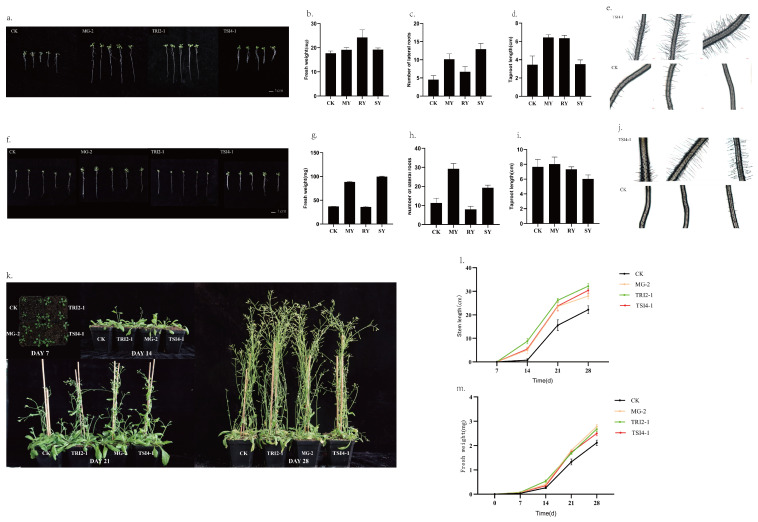
Effects of strains MG-2, TRI2-1, or TSI4-1 on the growth of *Arabidopsis*: (**a**–**e**) the growth and related index statistics of *Arabidopsis* treated with MG-2, TRI2-1, or TSI4-1 for 7 days; (**f**–**j**) the growth and related index statistics of *Arabidopsis* treated with MG-2, TRI2-1, or TSI4-1 for 14 days; (**k**–**m**) the growth of *Arabidopsis* and related growth index statistics after different root irrigation treatments of MG-2, TRI2-1, or TSI4-1. CK refers to the control group treated with sterile water, MY refers to the experimental group treated with strain MG-2, RY refers to the experimental group treated with strain TRI2-1, and SY refers to the experimental group treated with strain TSI4-1.

**Figure 9 microorganisms-11-01645-f009:**
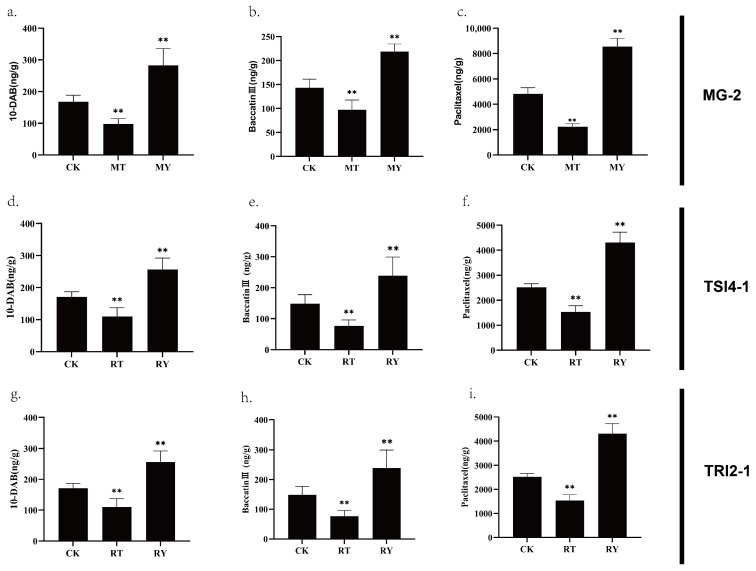
The content of paclitaxel and its precursors in *Taxus yunnanensis* stem cells after different treatments of MG-2, TRI2-1, or TSI4-1: (**a**–**c**) the content of paclitaxel and its precursor in stem cells after treatment with *Sphingomonas* sp. strain MG-2; (**d**–**f**) the content of paclitaxel and its precursor in stem cells after treatment with *Micromonospora* sp. strain TSI4-1; (**g**–**i**) the content of paclitaxel and its precursor in stem cells after treatment with *Kocuria* sp. strain TRI2-1. MY, the experimental group treated by the filtrate of the bacterial liquid of the strain MG-2; MT, the experimental group treated by the inactivated thallus of strain MG-2; RY, the experimental group treated with the filtrate of strain TRI2-1; RT, the experimental group treated by the inactivated thallus of strain TRI2-1; SY, the experimental group treated by the bacterial liquid filtrate of strain TSI4-1; ST, the experimental group treated by the inactivated thallus of strain TSI4-1. ** Represent statistically significant differences in comparison.

**Table 1 microorganisms-11-01645-t001:** Alpha Indices Statistics.

Sample Name	Observed_Species	Shannon	Simpson	Chao 1	ACE	Goods_Coverage	PD_Whole_Tree
Endophytic bacteria							
TR.1	442	4.432	0.827	569.317	591.599	0.988	33.877
TR.2	577	5.22	0.893	825.061	852.577	0.981	45.092
TR.3	535	4.749	0.805	679.222	713.189	0.985	42.331
TL.1	28	0.157	0.029	31	33.452	0.999	3.373
TL.2	31	0.128	0.025	146.5	103.173	0.998	4.162
TL.3	83	0.257	0.043	170.5	186.929	0.996	8.284
TS.1	47	0.202	0.034	68.375	71.53	0.998	4.637
TS.2	24	0.111	0.019	33	40.664	0.999	2.488
TS.3	39	0.191	0.033	47.667	52.891	0.999	4.261
Endophytic fungi							
TR.1	497	3.877	0.867	559.129	570.758	0.999	328.138
TR.2	315	2.629	0.681	336	353.881	0.999	196.471
TR.3	702	4.183	0.866	780.705	790.063	0.998	421.522
TL.1	761	4.437	0.9	833.02	851.161	0.998	360.4
TL.2	1017	5.169	0.92	1328.675	1236.798	0.996	534.092
TL.3	457	2.712	0.686	512.152	530.888	0.999	236.541
TS.1	937	4.622	0.871	1021.037	1035.52	0.998	450.165
TS.2	326	3.261	0.785	368.981	373.437	0.999	194.626
TS.3	394	3.15	0.803	445.891	448.959	0.999	210.703

Observed_species: the number of species observed (the number of OTUs). Shannon: The total number of categories in the sample and its proportion. The higher the community diversity, the more uniform the species distribution and the greater the Shannon index. Simpson: The diversity and evenness of species distribution in the community. This analysis uses Simpson‘s Index of Diversity (1-D). Chao 1: estimate the total number of species contained in community samples. ACE: estimate the number of OTUs in the community. Goods_coverage: sequencing depth index. PD_whole_tree: the genetic relationship of species in the community. TR: root. TL: leaf. TS: stem.

## Data Availability

Data may be provided at a reasonable request to the corresponding author. For culture-free sequence data, the accession numbers SUB13500522 and PRJNA980435 are recommended, while the accession number SUB13500932 is recommended for pure culture sequences.

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
