# Peer review of "Diversity of Endophytic Microbes in Taxus yunnanensis and Their Potential for Plant Growth Promotion and Taxane Accumulation"

_microorganisms, 2023, doi:10.3390/microorganisms11071645_

Round 1
Reviewer 1 Report
Dear authors,
The manuscript microorganism 2425293 “Diversity of endophytic microbes in Taxus yunnanensis and their potential for plant growth promotion and taxanes accumulation” includes the isolation of bacteria and fungi endophytes to help to accumulate several compounds related with Taxol production. The authors need improve the manuscript and considered the following comments.
Specific comments
1. Line 25: Why these strains are considered novel?
2. Please in all text the word “Arabidopsis” must be written in italics, because is a scientific name of the plant.
3. In paragraph 2.6 How many bacteria was tested.
4. In the figure 1, the bacteria and fungi names must be written in italics and the figure quality must be improve.
5. The quality of Figures 2 and 5 must be improved.
6. In the methodology paragraph. The authors need explain how they developed the different assays. For example, is important mentioned how the strain identification was done.
7. In phylogenetic tree the bacteria names must be written in italics. Also, the accession number given.
Minor points
Abstract
· Line 19. “in vitro” must be written in italics.
· Line 22: Changed “identification” by “selected”
Introduction
· Line 57: The correct word is “deaminase”, please changed “dehydrogenase” by “deaminase”.
· Line 72: The word “Niger” must be written without capital letters.
Material and methods.
Line 91: The correct word is “deaminase”, please changed “dehydrogenase” by “deaminase”.
Line 149: Changed “invoked” by “considered”.
Results
Line 219: Changed “ample” by “sample”.
Discussion
Line 537: Please changed “actinomyces” by “actinobacteria”
Only some mistakes need to be changed.
Author Response
Dear reviewer,
We would like to express our heartfelt gratitude for your meticulous review of our manuscript. Thank you for pointing out the details of the writing. We have answered or revised all the questions point-to-point. The modified areas have been displayed in red. Thank you again for your efforts.
Specific comments
- Line 25: Why these strains are considered novel?
Answer: This is an expression error, the ‘novel strains’ in line 25 means ‘Kocuria sp. TRI2-1, Micromonospora sp. TSI4-1, and Sphingomonas sp. MG-2’, it doesn't mean these are new strains. As per the reviewer's suggestion, we have refined the language of the paragraph in question to comply with academic norms and standards.
- Please in all text the word “Arabidopsis” must be written in italics, because is a scientific name of the plant.
Answer: We express our deepest gratitude to the reviewer for bringing this issue to our attention, and we have taken prompt action to address it. All concerns raised during the review process have been thoroughly examined, and any necessary corrections or improvements have been made accordingly.
- In paragraph 2.6 How many bacteria was tested.
Answer: We wish to convey our appreciation to the reviewer for their valuable feedback on our manuscript. We acknowledge that our explanation in the materials and methods section on this matter was not as clear and comprehensive as it should have been. Based on the comprehensive findings from our previous growth-promoting characteristics and plant experiments, we selected Kocuria sp. TRI2-1, Micromonospora sp. TSI4-1, and Sphingomonas sp. MG-2 for the induction experiment of T. yunnanensis stem cells.
- In the figure 1, the bacteria and fungi names must be written in italics and the figure quality must be improve.
Answer: Thank you for pointing out the shortcomings in the drawing process. We have made the necessary modifications as required.
- The quality of Figures 2 and 5 must be improved.
Answer: Thank you for pointing out the shortcomings in the drawing process. The quality of the pictures has all been improved.
- In the methodology paragraph. The authors need explain how they developed the different assays. For example, is important mentioned how the strain identification was done.
Answer: We appreciate the reviewer's efforts to review our work thoroughly and provide constructive feedback. Based on your comments, we have added supplementary information to clarify the relevant method in the corresponding section of our paper.
- In phylogenetic tree the bacteria names must be written in italics. Also, the accession number given.
Answer: We would like to express our sincere gratitude to the reviewer for their careful review of our manuscript. We appreciate their valuable feedback regarding the shortcomings in the drawing process and have proceeded to make the requisite modifications. Specifically, we have ensured that the bacterial names are written in italics in compliance with academic conventions and have included accession numbers for identification and verification purposes.
Minor points
Abstract
- Line 19. “in vitro” must be written in italics.
Answer: Thanks. We have made the necessary corrections.
- Line 22: Changed “identification” by “selected”
Answer: Thanks. We have made the necessary corrections.
Introduction
- Line 57: The correct word is “deaminase”, please changed “dehydrogenase” by “deaminase”.
Answer: Thanks. We have made the necessary corrections.
- Line 72: The word “Niger” must be written without capital letters.
Answer: Thanks. We have made the necessary corrections.
Material and methods.
Line 91: The correct word is “deaminase”, please changed “dehydrogenase” by “deaminase”.
Answer: Thanks. We have made the necessary corrections.
Line 149: Changed “invoked” by “considered”.
Answer: Thanks. We have made the necessary corrections.
Results
Line 219: Changed “ample” by “sample”.
Answer: Thanks. We have made the necessary corrections.
Discussion
Line 537: Please changed “actinomyces” by “actinobacteria”
Answer: Thanks. We have made the necessary corrections.
Reviewer 2 Report
Which bioinformatic pipeline was used for the analysis of the sequences? Add this information in the material and methods
Sample collection is not well described in the materials and methods: it should be mentioned which plant tissues were actually collected. LefSe analysis is not well described in the materials and methods: it is not clear whether it was performed at OTU level or not and it is not clear to me why the authors did not perform also the comparison between leaves and stem (TL vs. TS).
Phylogenetic analysis is not described in the materials and methods. Moreover, the phylogenetic tree (Fig. 7) should be improved for style: scientific names must be written in italics and bootstrap values at banches should be moved for better visibility.
Accession numbers of newly produced sequences (both cultivation-independent and isolates) must be provided.
For the plant experiments, ANOVA and post-hoc test should be used instead of Student's T-test, because there are three or more groups to compare.
In the supplementary material, the legends should be placed below each corresponding figure.
Author Response
Dear reviewer,
We would like to express our heartfelt gratitude for your meticulous review of our manuscript. Thank you for pointing out the details of the writing. We have answered or revised all the questions point-to-point. The modified areas have been displayed in red. Thank you again for your efforts.
Which bioinformatic pipeline was used for the analysis of the sequences? Add this information in the material and methods
Answer: We appreciate the valuable feedback provided by the reviewer regarding our materials and methods section. We recognize that there were some shortcomings in the description of the bioinformatics analysis process, and we apologize for any confusion this may have caused. To address this gap, we have added further information and details related to the analysis methodology in the relevant sections of the materials and methods.
Sample collection is not well described in the materials and methods: it should be mentioned which plant tissues were actually collected. LefSe analysis is not well described in the materials and methods: it is not clear whether it was performed at OTU level or not and it is not clear to me why the authors did not perform also the comparison between leaves and stem (TL vs. TS).
Answer: We appreciate the reviewer's valuable comments on our research. The relevant portions of the materials and methods section related to the collection of materials have been revised accordingly to improve clarity and accuracy. Regarding the LefSe analysis, we wish to clarify that the OUTs utilized in the analysis were normalized OTUs. The normalization method has been added to the corresponding section of the materials and methods to provide clarity on this matter.We performed LefSe analysis on the samples derived from both the stems and leaves of the studied plants. However, the observed differences in OUTs between these two sample types were relatively small. Thus, we did not find it necessary to report any specific results related to this comparison in our study.
Phylogenetic analysis is not described in the materials and methods. Moreover, the phylogenetic tree (Fig. 7) should be improved for style: scientific names must be written in italics and bootstrap values at banches should be moved for better visibility.
Answer: We express our sincere gratitude to the reviewer for their invaluable feedback on our manuscript. We greatly appreciate their insights into the phylogenetic tree presented in Figure 7, and we have taken prompt action to address the concerns raised. Specifically, we have made the necessary modifications to the tree based on the reviewer's suggestions to ensure that it accurately reflects the data presented in our study.
Accession numbers of newly produced sequences (both cultivation-independent and isolates) must be provided.
Answer: We would like to express our gratitude to the reviewer for raising an important question. In response, we have uploaded all sequences corresponding to the free culture and pure culture to the NCBI database. Furthermore, access numbers for these sequences have been included for ease of reference.
For the plant experiments, ANOVA and post-hoc test should be used instead of Student's T-test, because there are three or more groups to compare.
Answer: We express our sincere gratitude to the reviewer for their thoughtful and thorough review of our manuscript. We appreciate their constructive feedback regarding the shortcomings in our data analysis, and we have taken the necessary steps to address these issues. Specifically, we have carefully reviewed our data analysis methodology and made improvements to ensure the accuracy and robustness of our findings.
In the supplementary material, the legends should be placed below each corresponding figure.
Answer: We extend our sincere appreciation to the reviewer for their insightful suggestions and recommendations on our manuscript. We have carefully considered their feedback and have made the necessary modifications to address the issues raised.
Reviewer 3 Report
This comprehensive study analyzed the potential of endophytic bacteria to stimulate the production of anticancer metabolites. The authors followed independent and dependent culturing approaches to explore the composition of microbial endophytes of leaves, stems, and roots of Taxus yunnanensis. They found that the most dominant genera were Pseudomonas for bacteria and Fusarium, among fungi. Inoculation of promising PGPB (with abilities to stimulate IAA production, cellulase, siderophore production, protease and ACC deaminase activity, inorganic phosphate-solubilization, and nitrogen fixation), showed growth promotion in A. thaliana. In stem cells of T. yunnanensis, bacterial powder of selected strains stimulated higher taxol, baccatin III, and 10-DAB content.
The MS is well organized, with materials and methods reproducible and statistical methods appropriate for the study. Although, there are several aspects that the authors should add, correct, or clarify to consider this study for publication.
Please see the comments below
Minor comments
L24. Not clear. Do you mean supernatant? according to M&M the authors used freeze-dryed bacteria; please, clarify
L96-L98. This sentence fits better in conclusions.
L118. Add the name of the database used for taxonomic annotation.
L131. One would expect that the ground sample is some king paste, not powder. Was there a special drying process? it does not seem necessary for endophytic isolation. Please revise.
L165. Indicate what was used in the standard curve.
L174. Add literature support for cellulase production test.
L195. Correct' surcose' to 'sucrose'
L198. It is not clear what 'stable stage' means. Please indicate Lag, exponential or stationary phase.
L202-L204. What was the idea of using the supernatant and the bacterial powder in the experiment with stem cell cultures?
L204. What do you mean by 'inducers'?
L238. Are these 29 strains of cultured bacteria? if not, they should be called OTUS
Table 1. Add the meaning for TR, TL, and TS.
Figure 7. Legend for figure: add the method of construction of the Phylogenetic tree.
L3982-L404. The comparison of lateral roots is not appropriate; see that Fig. 8 shows different sections of the roots. Please revise.
L511. The predicted 'results' or 'functions'? and how?
Minor corrections in the English language required
Author Response
Dear reviewer,
We would like to express our heartfelt gratitude for your meticulous review of our manuscript. Thank you for pointing out the details of the writing. We have answered or revised all the questions point-to-point. Modifications have been made to the corresponding positions in the manuscript. Thank you again for your efforts.
L24. Not clear. Do you mean supernatant? according to M&M the authors used freeze-dryed bacteria; please, clarify
Answer: We appreciate the reviewer's efforts to carefully review our work and provide valuable feedback. We have conducted experiments on both the supernatant and the bacterial body, as indicated in our methods section, and have found that the supernatant has a better promoting effect on the accumulation of taxanes in Taxus yunnanensis stem cells. As a result, the abstract emphasizes the results of the experiment conducted on the supernatant specifically.
L96-L98. This sentence fits better in conclusions.
Answer: We would like to express our sincere gratitude to the reviewer for their insightful suggestions and recommendations, which have helped to enhance the quality and clarity of our manuscript. We have carefully considered your feedback and have made the appropriate modifications to our work accordingly.
L118. Add the name of the database used for taxonomic annotation.
Answer: We extend our sincere appreciation to the reviewer for their careful review of our manuscript and their insightful questions regarding our research methodology. We have taken their feedback into consideration and have made the necessary additions to the materials and methods section of our manuscript in order to provide a more comprehensive description of our research protocol.
L131. One would expect that the ground sample is some king paste, not powder. Was there a special drying process? it does not seem necessary for endophytic isolation. Please revise.
Answer: We would like to express our gratitude to the reviewer for their insightful questions and suggestions regarding our research methodology. The ground sample is king paste, not powder. We have corrected.
L165. Indicate what was used in the standard curve.
Answer: We would like to acknowledge the reviewer for raising pertinent questions. As a result, we have incorporated the necessary supplements in the materials and methods section of our research.
L174. Add literature support for cellulase production test.
Answer: We appreciate your comments regarding the references and have taken steps to supplement them accordingly.We understand the importance of adhering to academic norms and appreciate your guidance in this regard. We have made the necessary changes to ensure that our manuscript meets the required standards.
L195. Correct' surcose' to 'sucrose'.
Answer: We apologize for the wording error that you pointed out and have made the necessary corrections to ensure the accuracy of our work.
L198. It is not clear what 'stable stage' means. Please indicate Lag, exponential or stationary phase.
Answer: Thank you for your valuable feedback. We appreciate your attention to detail and have made the necessary revisions to clarify the term 'stable stage' in our paper. As per your suggestion, we have supplemented the corresponding section to explicitly state that the stable stage refers to the stationary phase.
L202-L204. What was the idea of using the supernatant and the bacterial powder in the experiment with stem cell cultures?
Answer: In this study, we hypothesize that both extracellular and intracellular substances may play a significant role in the accumulation of taxanes in Taxus yunnanensis stem cells. To test this hypothesis, we have designed experiments to investigate the effects of both cell and fermentation broth on taxane accumulation. It is well-established that taxanes are produced and stored within the cells of Yunnan yew. However, recent studies have suggested that extracellular substances, such as plant growth regulators and elicitors, may also influence taxane production. Therefore, we believe that investigating the role of both extracellular and intracellular substances is necessary to gain a comprehensive understanding of taxane accumulation in Taxus yunnanensis stem cells. To achieve this, we have conducted experiments on both the cell and fermentation broth. We have analyzed the taxane content in the cells to determine the contribution of each component to taxane accumulation.
L204. What do you mean by 'inducers'?
Answer: In the present study, the term "inducers" pertains to both the supernatant and bacterial powder, which were incorporated into the culture medium of Taxus yunnanensis stem cells in minimal quantities. The use of inducers in this context is a well-established practice in cell culture research, and is based on the principle of introducing external factors that can modulate the behavior of the cells in a controlled manner.
L238. Are these 29 strains of cultured bacteria? if not, they should be called OTUS.
Answer: Thanks. These strains are not of cultured bacteria, so we have changed them to OUTS.
Table 1. Add the meaning for TR, TL, and TS.
Answer: Thank you for your valuable feedback. The meaning for TR, TL, and TS has been added to the note of Table 1.
Figure 7. Legend for figure: add the method of construction of the Phylogenetic tree.
Answer: We would like to express our gratitude to the reviewer for bringing to our attention the limitations of our study. In response to the feedback, we have made the necessary revisions to improve the clarity and accuracy of our findings. Specifically, we have included the drawing method of the phylogenetic tree in the annotations of Figure 7 to provide a more comprehensive understanding of our results.
L3982-L404. The comparison of lateral roots is not appropriate; see that Fig. 8 shows different sections of the roots. Please revise.
Answer: We express our gratitude to the reviewer for bringing to our attention the irregularities in the images. We have taken the necessary steps to rectify the issues, and have significantly enhanced the quality of the images.
L511. The predicted 'results' or 'functions'? and how?
Answer: Thanks. It should be functions, and we inferred this conclusion based on the predicted functions that the presence of numerous endophytic bacteria may be capable of promoting plant growth and enhancing the synthesis of plant secondary metabolites.